# Chemonucleolysis with Chondroitin Sulfate ABC Endolyase for Treating Lumbar Disc Herniation: Exploration of Prognostic Factors for Good or Poor Clinical Outcomes

**DOI:** 10.3390/medicina56110627

**Published:** 2020-11-19

**Authors:** Katsuhiko Ishibashi, Muneyoshi Fujita, Yuichi Takano, Hiroki Iwai, Hirohiko Inanami, Hisashi Koga

**Affiliations:** 1Department of Orthopaedics, Iwai FESS Clinic, 8-18-4 Minamikoiwa Edogawa-ku, Tokyo 133-0056, Japan; muneyoshi.fujita.0302@main.teikyo-u.ac.jp (M.F.); h-iwai@iwai.com (H.I.); ina@iwai.com (H.I.); hkoga0808@gmail.com (H.K.); 2Department of Orthopaedics, Iwai Orthopaedic Medical Hospital, 8-17-2 Minamikoiwa Edogawa-ku, Tokyo 133-0056, Japan; y-takano@iwai.com; 3Department of Orthopaedic Surgery, Teikyo University School of Medicine, 2-11-1 Kaga, Itabashi-ku, Tokyo 173-8606, Japan; 4Department of Orthopaedic Surgery, Inanami Spine and Joint Hospital, 3-17-5 Higashishinagawa Shinagawa-ku, Tokyo 140-0002, Japan

**Keywords:** chemonucleolysis, condoliase, chondroitin sulfate ABC endolyase, lumber disc herniation

## Abstract

*Background and Objectives:* Chondroitin sulfate ABC endolyase (condoliase) was launched as a new drug for chemonucleolysis in 2018. Few studies assessed its clinical outcomes, and many important factors remain unclear. This study aimed to clarify the preoperative conditions in which condoliase could be highly effective. *Materials and Methods:* Of 47 patients who received condoliase, 34 were enrolled in this study. The mean age of the patients was 33 years. The average duration since the onset of disease was 8.6 months. We evaluated patients’ low back and leg pain using a numerical rating scale (NRS) score at two time points (before therapy and 3 months after therapy). We divided the patients into two groups (good group (G): NRS score improvement ≥ 50%, poor group (P): NRS score improvement < 50%). The parameters evaluated were age, disease duration, body mass index (BMI), and positive or negative straight leg raising test results. In addition, the loss of disc height and preoperative radiological findings were evaluated. *Results:* In terms of low back and leg pain, the G group included 9/34 (26.5%) and 21/34 (61.8%) patients, respectively. Patients’ age (low back pain G/P, 21/36.5 years) was significantly lower in the G group for low back pain (*p* = 0.001). High-intensity change in the protruded nucleus pulposus (NP) and spinal canal occupancy by the NP ≥ 40% were significantly high in those with leg pain in the G groups (14/21, *p* = 0.04; and 13/21, *p* = 0.03, respectively). *Conclusions:* The efficacy of improvement in leg pain was significantly correlated with high-intensity change and size of the protruded NP. Condoliase was not significantly effective for low back pain but could have an effect on younger patients.

## 1. Introduction

Chemonucleolysis has been considered a minimally invasive treatment for cervical and lumbar disc herniation (LDH) for more than 50 years [1]. Although several enzymes (trypsin, hyaluronidase, cathepsin G, chymotrypsin, and calpain) have been proposed as therapeutic agents [2,3,4,5,6], only two enzymes (chymopapain and collagenase) have been used in clinical practice [1,7]. Chymopapain was introduced in July 1963 and was widely used throughout Europe, North America, and Australia [8,9,10]. Following reports of severe adverse events, including fatal anaphylaxis, bleeding, and neurologic complications [11,12,13,14,15,16], the production of chymopapain (Chymodiactin^®^, Smith Laboratories Inc., Maple Grove, MN, USA) was discontinued for nonscientific and commercial reasons in 2002. The first clinical study on collagenase (nucleolysin) was reported in 1981 [7]. Subsequently, collagenase was reported to be less injurious to the spinal nerve roots and perineural tissues; however, it was reported to be less clinically effective than chymopapain [17]. Severe adverse events were also associated with this enzyme (cauda equina syndrome and disc herniation due to “digestion” of the annulus fibrosus) [8,18,19].

A safer and more effective chemonucleolysis enzyme was necessary, and chondroitin sulfate ABC endolyase (condoliase) from *Proteus vulgaris* was developed. To date, several basic experimental studies have shown promising results [12,20,21,22,23,24,25], and condoliase was finally launched in August 2018 in the Japanese clinical field. The target of condoliase is chondroitin sulfate, which is distributed in the nucleus pulposus (NP) but not in nerves and vascular tissues. Therefore, condoliase can be safely and specifically used for the treatment of LDH.

Phase 1–3 clinical trials of condoliase were conducted in Japan. Matsuyama et al. reported the results of phases 2 and 3 trials of chemonucleolysis for the treatment of LDH (Clinical trial registration no. NCT00634946) [11]. In this study, 194 patients received three different doses of condoliase (1.25, 2.5, or 5 U) or placebo injections, and the appropriate dose of injection was determined (1.25 U). Chiba et al. reported the results of phase 3 trials about chemonucleolysis with condoliase for LDH treatment [26]. In this study, 163 patients received 1.25 U of condoliase or placebo injections, and the leg pain significantly improved at 52 weeks after condoliase injection. Considering these good outcomes, chemonucleolysis with condoliase is being used to treat patients of LDH with radiculopathy who are resistant to conventional conservative treatments.

Here, we report our therapeutic experience of 34 cases that were available for follow-up for more than 3 months out of 47 cases treated with condoliase in our hospital from August 2018 to August 2019.

## 2. Materials and Methods

### 2.1. Patient Selection

LDH patients complaining of low back and/or leg pain were conservatively treated with medications and block therapy for at least 1 month. Chemonucleolysis with condoliase was considered in patients in whom the unilateral leg pain did not improve even after 1 month and in patients who did not wish to undergo surgery.

The inclusion criteria were as follows: (1) LDH of the extrusion and subligamentous extrusion types, in which spontaneous regression is difficult; and (2) the transligamentous extrusion type of LDH in which only a part of the posterior longitudinal ligament (PLL) is ruptured. Chemonucleolysis with condoliase was performed in 47 patients (34 men, 13 women) in our hospital from August 2018 to October 2019. We could include only 34 patients (24 men, 10 women) who could be followed-up for 3 months or more without any additional intervention (Figure 1).

In the early stage of this study, chemonucleolysis with condoliase was performed on an intraforaminal LDH with spondylolisthesis in a 68-year-old woman; however, the leg pain worsened, and full-endoscopic spine surgery (FESS) was required 3 months after chemonucleolysis. Subsequently, we excluded patients with spondylolisthesis (defined as ≥3 mm vertebral slipping based on the lateral-lumbar radiograph) and/or with instability (posterior intervertebral angle ≥ 5°). 

Among the 13 patients who dropped out, two underwent surgical treatment within 3 months after chemonucleolysis owing to poor pain relief. The remaining 11 patients could not be evaluated using postoperative plain radiographs or magnetic resonance imaging (MRI) because they did not visit our outpatient clinic.

### 2.2. Procedure

Patients were carefully log rolled into the prone position. During the procedure, a fluoroscope was placed across the center of the operating table to ensure appropriate positioning of the needle. In case of restricted fluoroscope movement, pelvic tilting was often performed to confirm the disc space and the needle trajectory (similar to the lateral recumbent position). After marking the site of entry on the skin, the skin was disinfected and covered with a surgical drape. Under local anesthesia, a 22-gauge spinal needle with a stylet was inserted into the center of the disc space using a posterolateral approach, from the side opposite to the side of radiculopathy. The position of the needle tip was confirmed with a posterolateral fluoroscopic view during insertion. The final position of the tip (the center of the NP) was confirmed by both anteroposterior and lateral fluoroscopic views. The use of contrast medium was disallowed during the entire process. The stylet was removed, and 1.25 U of condoliase (HERNICORE^®^, Seikagaku Corporation, Tokyo, Japan) dissolved in 1.2 mL of saline was injected. To ensure a prompt response in case of unexpected adverse events such as anaphylaxis, an intravenous line was established prior to the injection, and an antibiotic was administered 30 min before the procedure. The patients were hospitalized on the day of the injection [27]. Four full-time surgeons at our hospital performed all the procedures; one of them was a spine specialist with 20 to 30 years of experience, and the remaining three were spine specialists with 10 to 20 years of experience.

### 2.3. Evaluation of the Patients’ Background

Evaluation of the numerical rating scale (NRS) score (10 indicates highest level of pain, 0 indicates no pain) was performed using a questionnaire that was filled by the patients themselves. The scores were evaluated before and 3 months after the procedure. The straight leg raising (SLR) test was performed in the outpatient department before the procedure. 

### 2.4. Image Evaluation

Plain radiographs and MRI were taken before and 1 month after the procedure and before and 3 months after the procedure, respectively.

#### 2.4.1. Height of the Intervertebral Disc

The change in the height of the intervertebral disc was evaluated from pre- and postoperative plain radiographs. We calculated the Brandner’s disc index [28] using the following 2 formulae: 

The anterior intervertebral disc height reduction rate is (B/A-B’/A’)/B/A × 100 (X is before the procedure X’ is after the procedure). The posterior intervertebral disc height reduction rate is (C/A-C’/A’)/C/A × 100 (X is before the procedure X’ is after the procedure). When we found at least one intervertebral disc height reduction rate ≥ 25%, we judged “disc height decreases ≥ 25%” (Figure 2). 

#### 2.4.2. Occupancy Ratio of the Spinal Canal by the Protruded Nucleus Pulposus (NP)

The occupancy ratio of the spinal canal by the protruded NP was measured using T2-weighted MRI on axial view. The areas of protruded NP and the corresponding spinal canal were calculated using an image measurement software. The area of the spinal canal was defined as the region enclosed by the original dorsal surface of the annulus fibrosus and the ventral edge of the ligamentum flavum. The area of the protruded NP was defined as the region enclosed by the original dorsal surface of the annulus fibrosus and the dorsal surface of the protruded NP. The occupancy ratio was calculated using the following formula: occupancy ratio of the spinal canal is the area of the protruded NP (blue line area)/the area of the spinal canal (red line) × 100 (%) (Figure 3).

#### 2.4.3. High-Intensity Change (HIC) of the Protruded Nucleus Pulposus (NP)

The HIC of the protruded NP was determined on the sagittal view of the T2-weighted MRI. We defined HIC of the protruded NP when the intensity of the protruded NP was higher than the intensity of the surface of the protruded NP (Figure 4).

### 2.5. Statistical Methods

We first evaluated whether each variable (age, disease duration, NRS score, and BMI) followed the normal distribution, using the Shapiro–Wilk normality tests. Age was found to be normally distributed, while disease duration, NRS score, and BMI were not normally distributed. For age, paired t-tests were performed as a parametric test. For the NRS score, Wilcoxon signed-rank test was performed as a nonparametric test. For disease duration and BMI, the Mann–Whitney U test was used as a nonparametric test. We divided the patients into two groups based on the improvement in NRS score (low back pain and leg pain). Good group (G): NRS score improvement ≥ 50%, poor group (P): NRS score improvement < 50%. Both groups were compared using the chi-square test or Fisher’s direct method. A *p*-value < 0.05 was considered statistically significant. All statistical analyses were performed using SPSS version 20.0 (SPSS Inc., Chicago, IL, USA). 

## 3. Results

The average age was 32.4 years (range, 13–68 years). The average duration since the onset of disease was 8.6 months (range, 1–26 months). There were 32 cases with intracanal LDHs, and 1 case each with foraminal LDH and extraforaminal LDH. Cases with LDH type (intracanal LDH) were as follows: protrusion in 5 patients, subligamentous extrusion in 24 patients, and trans-ligamentous extrusion in 3 patients. The LDH level was L4/5 in 25 patients and L5/S in 9 patients. Except for one patient with recurrence, the remaining 33 patients had de novo LDH (Table 1).

We divided the patients into two groups (good group (G): NRS score improvement ≥ 50%, poor group (P): NRS score improvement < 50%). In terms of low back and leg pain, the G group included 9/34 (26.5%) and 21/34 (61.8%) patients, respectively. The NRS score for leg pain improved significantly 3 months after chemonucleolysis with condoliase (Figure 5). Regarding age (low back pain G/P 21/36.4 years), low back pain in the G group was seen more in the younger age group compared to that in the P group (*p* = 0.001) (Table 2). Regarding disease duration (low back pain G/P 7.4/9.3 months, leg pain G/P 8.4/9.1 months) and BMI (low back pain G/P 22.4/23.9, leg pain G/P 23.5/21.7), there was no significant difference between the two groups (Table 2 and Table 3). Furthermore, there was no significant difference between the two groups in the proportion of patients who had a positive SLR test. 

The preoperative imaging findings were not significantly different between the two groups in patients with low back pain (Table 4). However, a significant difference was observed between the two groups in patients with leg pain with respect to the proportion of patients with HIC of the protruded NP (14/21 in G, 4/13 in P, *p* = 0.04) (Table 5). A significant difference was also observed between patients with leg pain in the two groups in terms of the spinal canal occupancy of NP. An occupancy ratio ≥ 40% was observed in the G group with leg pain (13/21 in G, 3/13 in P, *p* = 0.03) (Table 5).

In some patients, low back pain increased within 1 month after chemonucleolysis; however, it usually improved within 1 month by taking analgesics. In one patient, low back pain worsened significantly after chemonucleolysis. The patient was a case of LDH with spondylolysis. We speculate that the decrease in the height of the intervertebral disc caused mechanical stress to the pars interarticularis defect, leading to the aggravation of low back pain (Figure 6).

## 4. Discussion

Chiba et al. reported a domestic phase 3 trial of condoliase for the treatment of protrusion type and sub-ligamentous extrusion type of LDH, in which an improvement in visual analog scale (VAS) score (improvement of VAS of leg pain ≥50% 12 weeks after administration of condoliase) was seen in 59/81 patients (72%) [26]. We used the NRS score for the evaluation of leg pain and defined improvement in NRS score ≥ 50% as good. This is similar to Chiba’s evaluation. The improvement rate in our study was 21/34 patients (61.8%). This was inferior to that in Chiba’s report. One of the reasons for this difference could be that Chiba’s study might have had a high rate of spontaneous remission, because a high rate of improvement (50%) was also observed in the placebo group. Banno et al. also reported that chemonucleolysis with condoliase was effective in 34/47 patients (70.2%); however, they defined a more than 20 mm decrease in the VAS score as effective. There is also a possibility of the inclusion of cases with trans-ligamentous extrusion that might have contributed to the good outcome in Table 3 of Banno’s report [30,31]. Comparing the data of previous studies and our study, at least 60% of the cases with extrusion and sub-ligamentous extrusion type of LDH might respond to chemonucleolysis.

From this study, it can be suggested that the size of the protruded NP might be a predictive factor for the improvement of leg pain. Chemonucleolysis with condoliase might be less effective in cases with protrusion type and the small subligamentous extrusion type; however, it is difficult to precisely distinguish between the two on MRI. Nevertheless, a spinal canal occupancy ratio ≥ 40% is highly predictive of improvement in leg pain. Various surgical approaches from decompression alone to fixation, have been reported for the treatment of large central LDHs, which are difficult to treat [32,33,34,35,36]. In recent years, transforaminal discectomy by FESS has been reported to have good results for large central LDHs [37,38,39]. However, chemonucleolysis with condoliase seems to be a less invasive treatment than FESS and is suitable for large central LDH. We show below, representative images of a patient with a large LDH treated by chemonucleolysis with condoliase in Figure 7. An 18-year-old man with L4/5 LDH complaining of low back pain and left leg pain was first treated conservatively for more than 12 months. There was no improvement, and chemonucleolysis was performed. The improvement in low back pain and leg pain was gradually seen 4 weeks after chemonucleolysis without any adverse events. The NRS scores for both, low back and leg pains improved significantly from 7 to 0 and from 3 to 0, respectively, at 6 months after the treatment (Figure 7).

Banno et al. investigated the signal intensity of the protruded NP and reported that patients with high intensity on T2-weighted MRI showed a better response to chemonucleolysis with condoliase [30]. From the results of our study as well, it can be suggested that high-intensity change (HIC) in the protruding NP on T2-weighted MRI might be another predictive factor for the improvement of leg pain. The combination of occupancy rate and HIC of the protruding NP might serve as patient selection criteria, to achieve a higher response to chemonucleolysis with condoliase.

We also obtained insightful information from patients who did not see an improvement or had aggravation of low back and/or leg pain after chemonucleolysis and underwent discectomy. Two patients underwent surgical intervention within 3 months and were excluded from the above analysis (foraminal LDH associated with spondylolisthesis and a patient with recurrent LDH after full-endoscopic discectomy). Six of the 34 patients underwent surgical intervention after 3 months of follow-up. Among the patients, eight required surgical intervention and two each had recurrent LDH, protrusion type LDH, and small sub-ligamentous extrusion type of LDH. The size of the protruded NPs did not change after chemonucleolysis. For recurrent LDH, scar formation following a previous surgery might affect the penetration of condoliase and shrinkage of the NP. The pre- and postoperative spinal canal occupancy ratio did not change in these eight patients. Therefore, recurrent LDH, protrusion type of LDH, and small sub-ligamentous extrusion type of LDH should be excluded from the indication for chemonucleolysis with condoliase. We also pathologically examined the removed disc material and confirmed the preservation of the cartilaginous endoplate. Furthermore, we compared pathologic tissues obtained from patients (32-year-old man) who underwent surgery 3 months after condoliase injection and the similar-aged patients (32-year-old man) in whom condoliase was not injected. Decrease of toluidine blue staining in the intercellular space of condoliase injection indicates the degradation of glycosaminoglycans (chondroitin sulfate, keratan sulfate, and hyaluronic acid), which is a main component of the nucleus pulposus. However, chondrocyte necrosis was scarcely observed in both specimens (Figure 8). This finding strongly supports the safety of condoliase against the surrounding normal tissues.

With respect to low back pain, Chiba et al. reported that there was no significant difference in the treatment group compared with the placebo group in terms of improvement at 3 months after administration [26]. In our study as well, there was no significant improvement 3 months after administration. However, comparison between the G and P groups with low back pain revealed that there was a significant difference in age. However, eight patients (four male, four female) in their teens were included in this study; the G group was statistically younger than the P group. Eight teenagers showed a good improvement of both low back pain and leg pain (mean improvement rate of NRS score of low back pain and leg pain ≥50% 3 months after the administration was 67.5% and 87.5%, respectively). We speculate that one of the reasons for the high efficacy in teenaged cases is the high water content in the nucleus pulposus (the content is predicted to be more abundant in younger patients than that in elder patients). Therefore, condoliase might be more effective in younger patients. The cause of low back pain is extremely complicated especially in elder patients, because the low back pain depends heavily on the other age-related factors (degeneration of disc, facet joint, muscle and ligaments, kyphotic change, and so on). On the other hand, the low back pain in younger patients might occur simply. In younger patients, the low back pain frequently improves after nerve root block, this indicates the cause of the low back pain is only compression of nerve roots by the protruding nucleus pulposus. We perceive that five of eight teenagers (67.5%) in this study had such a pathophysiological status. Chemonucleolysis with condoliase should be carefully considered for younger patients (<20 years) because of their growth and development. Since we observed no adverse events in the abovementioned eight teenagers during the short follow-up period (range: 10–21 months), the indication of chemonucleolysis with condoliase might be extended to include younger patients.

### Limitation of This Study

The limitations of this study are its retrospective nature, a low number of patients, an unequal distribution between males and females, and short follow-up period.

## 5. Conclusions

This was a study of 34 patients with LDH, who underwent chemonucleolysis with condoliase. It was found that chemonucleolysis with condoliase was more effective for leg pain than for low back pain. Preoperative high signal intensity of the protruded NP on T2-weighted MRI and large protruded NP (occupancy ratio in spinal canal ≥ 40%) were potential predictive factors for condoliase response. Although chemonucleolysis with condoliase was not significantly effective for low back pain, it might be more effective for low back pain in the younger population (<20 years). It is concluded that chemonucleolysis with condoliase is a safe and effective treatment for LDH that is resistant to conservative therapy. It is expected that the indications will be expanded to juvenile LDH and other vertebral areas in the future.

## Figures and Tables

**Figure 1 medicina-56-00627-f001:**
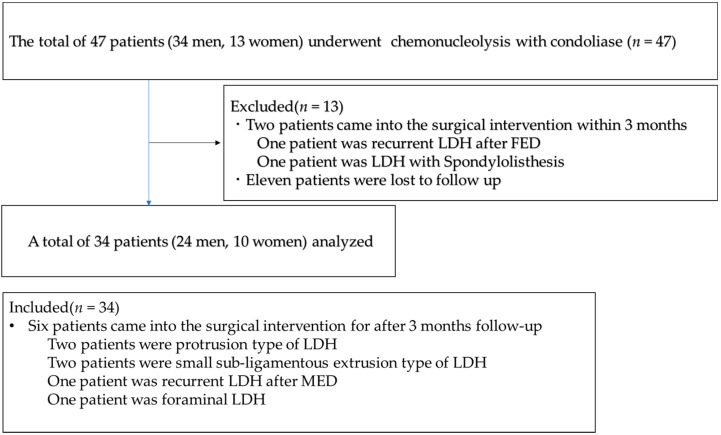
Flow diagram of this study. LDH, lumbar disc herniation; FED, full-endoscopic discectomy; MED, micro-endoscopic discectomy.

**Figure 2 medicina-56-00627-f002:**
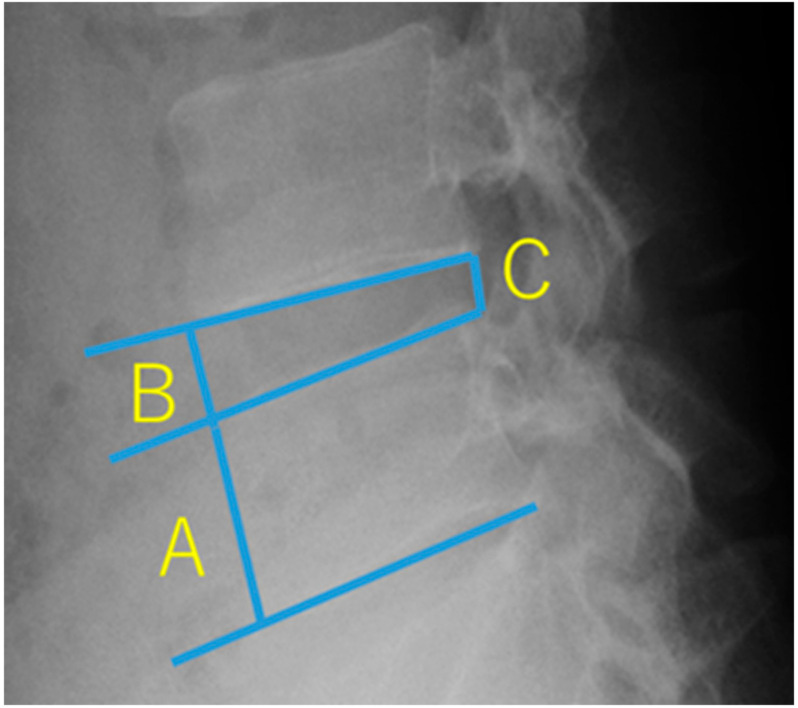
Brandner’s disc index. A is the maximum height of the vertebral body. B and C are the disc heights at the anterior and posterior portions, respectively. B/A is the anterior disc index. C/A is the posterior disc index.

**Figure 3 medicina-56-00627-f003:**
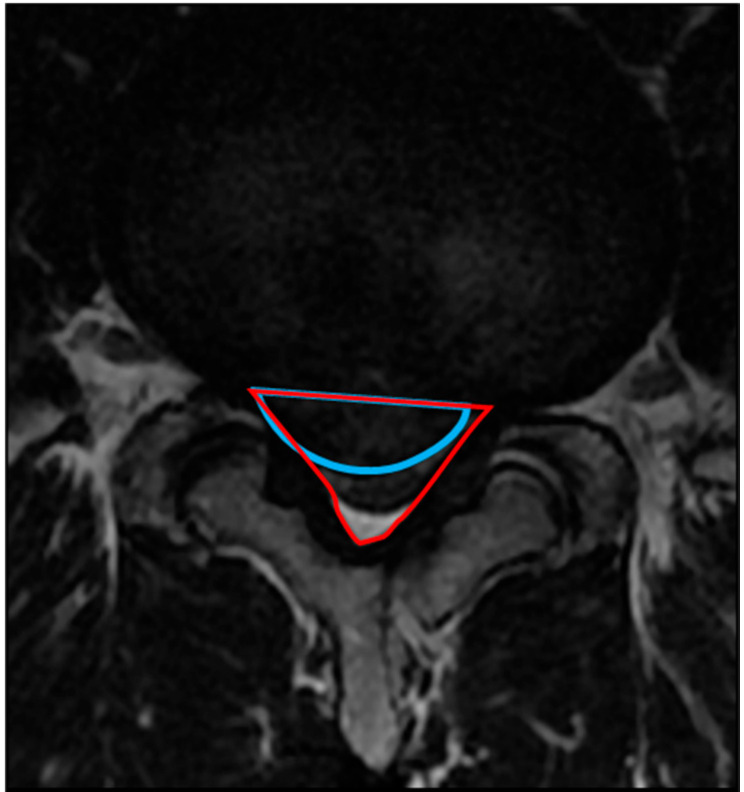
Method for calculating the occupancy ratio of spinal canal by the protruded nucleus pulposus (NP). The area of the protruded NP is the region enclosed by the blue arc. The area of the spinal canal is the region enclosed by the red triangle.

**Figure 4 medicina-56-00627-f004:**
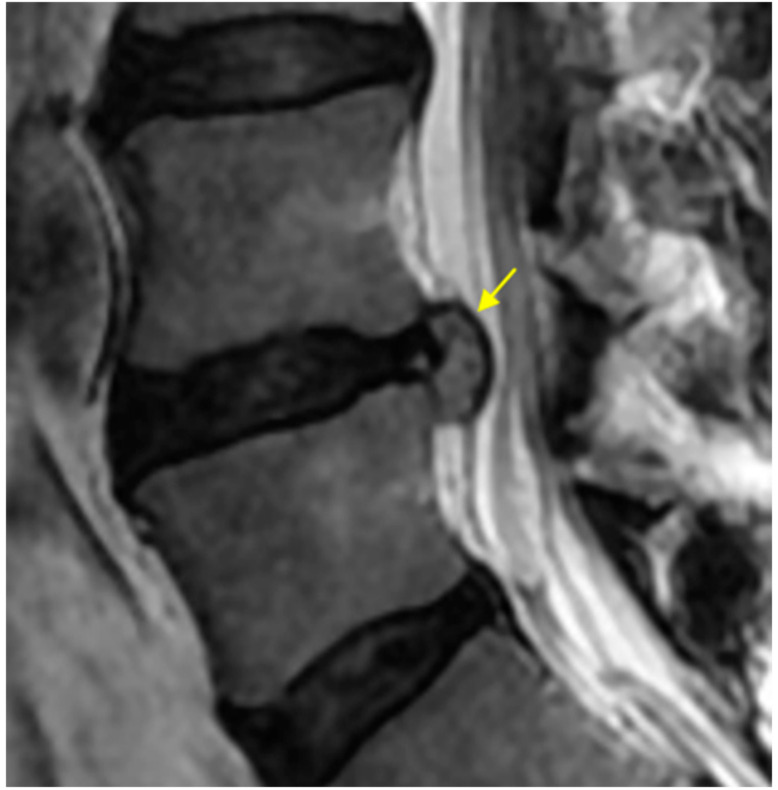
High intensity change of protruded nucleus pulposus. The intensity of the protruded NP content is higher than that of the protruded NP surface (yellow arrow).

**Figure 5 medicina-56-00627-f005:**
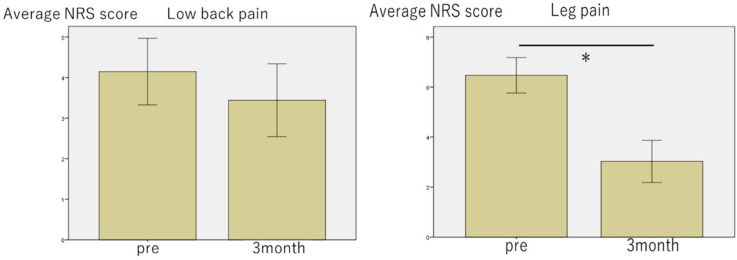
Bar chart of changes in the numerical rating scale (NRS) score for low back pain and leg pain following chemonucleolysis with condoliase. The NRS score for leg pain improved significantly 3 months after the injection. * *p* < 0.05 on Wilcoxon signed-rank test.

**Figure 6 medicina-56-00627-f006:**
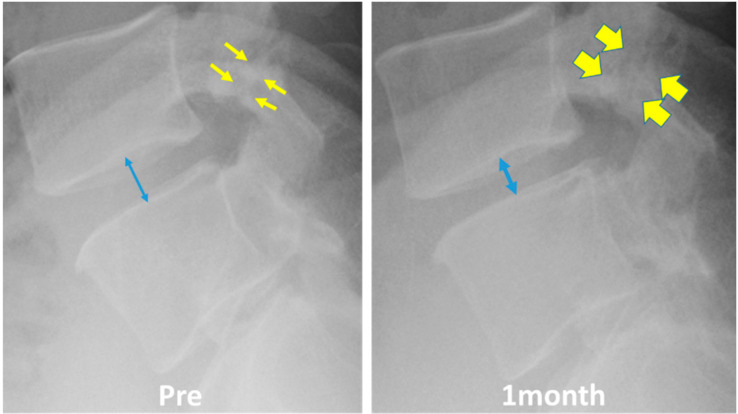
A patient of herniation with spondylolysis in whom low back pain worsened significantly after chemonucleolysis. The decrease in the intervertebral disc height (blue arrow) might cause mechanical stress to the pars interarticularis defect (yellow arrows).

**Figure 7 medicina-56-00627-f007:**
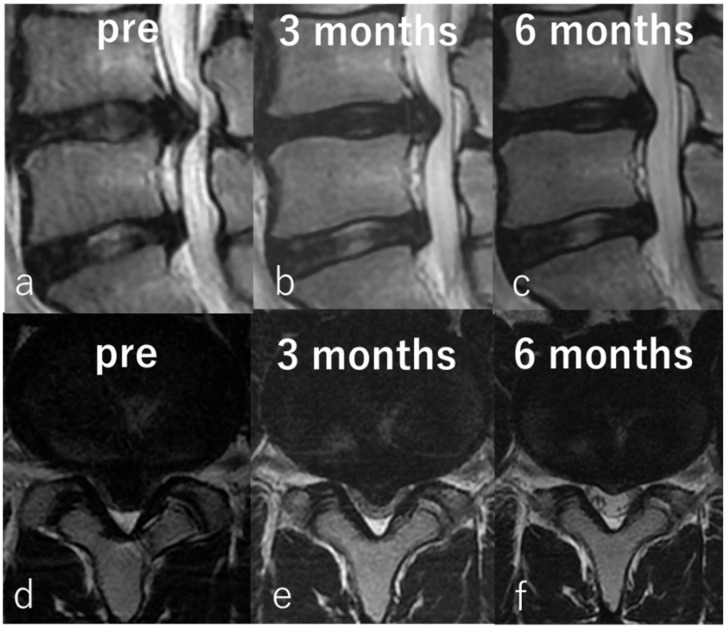
Representative images of an 18-year-old man with large LDH. Preoperative sagittal (**a**) and axial (**d**) T2-weighted magnetic resonance imaging (MRI) showing sub-ligamentous LDH at L4/5. Sagittal (**b**) and axial (**e**) MRI taken 3 months after chemonucleolysis showing reduction of LDH. Sagittal (**c**) and axial (**f**) MRI taken 6 months after chemonucleolysis showing significant reduction in size.

**Figure 8 medicina-56-00627-f008:**
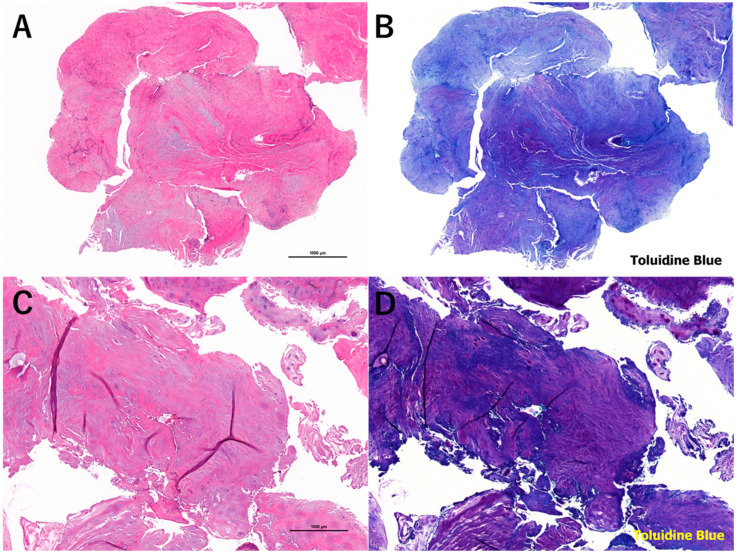
Micrographs of the nucleus pulposus obtained from patients who underwent surgery 3 months after condoliase injection (**A**,**B**) and the similar-aged patients in whom condoliase was not injected (**C**,**D**). Staining: (**A**,**C**) hematoxylin and eosin; (**B**,**D**) toluidine blue. Magnification of microscope: (**A**–**D**) ×20.

**Table 1 medicina-56-00627-t001:** Demographic data of 34 patients.

All Patients	(*n* = 34)
Age (years)	32.4 ± 13.0
Female	10
BMI	22.8 ± 4.07
Herniation level	
L4/5	25
L5/S	9
disease duration (months)	8.6 ± 6.54
SLR	
positive	20
negative	10
data loss	4
NRS for back pain	4.15 ± 2.35
NRS for leg pain	6.47 ± 2.03
Herniation type	
protrusion	5
sub-ligamentous	24
trans-ligamentous	3
Intraforaminal herniation	1
Extraforaminal herniation	1
Pfirrmann classification	
Grade II	13
Grade III	20
Grade IV	1
Postoperative recurrent herniation	1
HIC of protruded NP on MRI	18

Continuous data are presented as the mean ± SD. Categorical data are presented as numbers. Disc degeneration was graded on T2-weighted MRI using a grading system proposed by Pfirrmann [29]. BMI, body mass index; SLR, straight leg raising; NRS, numerical rating scale; LDH, lumbar disc herniation; HIC, high-intensity change; NP, nucleus pulposus; MRI, magnetic resonance imaging.

**Table 2 medicina-56-00627-t002:** Comparison of demographic and baseline characteristics of patients with low back pain between the good (G) and poor (P) groups.

Low Back Pain	Group G (*n* = 9)	Group P (*n* = 25)	*p-*Value
Age (year)	21.0 ± 6.16	36.4 ± 12.4	**0.001 ***
disease duration (month)	7.4	9.3	0.344 **
BMI	23.9	22.4	0.612 **
SLR positive	6	14	1.00 ^†^

*: paired *t*-test, **: Mann–Whitney U test, ^†^: χ^2^ test. *p* < 0.05 is shown in bold.

**Table 3 medicina-56-00627-t003:** Comparison of demographic and baseline characteristics of patients with leg pain between the G and P groups.

Leg Pain	Group G (*n* = 21)	Group P (*n* = 13)	*p-*Value
Age (year)	30.3 ± 14.3	35.6 ± 10.2	0.257 *
disease duration (month)	8.4	9.1	0.531 **
BMI	23.5	21.7	0.523 **
SLR positive	14	6	1.000 ^†^

*: paired *t*-test, **: Mann–Whitney U test, ^†^: χ^2^ test.

**Table 4 medicina-56-00627-t004:** Comparison of imaging findings of patients with low back pain between the G and P groups.

Low Back Pain	Group G (*n* = 9)	Group P (*n* = 25)	*p*-Value
disc height decreases ≥ 25%	5	16	0.704 ^†^
HIC of protruded NP on MRI	5	13	1.000 ^†^
the spinal canal occupying ratio ≥ 40%	4	12	1.000 ^†^
Pfirrmann classification			0.381 ^†^
Grade II	2	10	
Grade III	7	13	
Grade IV	0	1	

^†^: χ^2^ test. HIC = high-intensity change, NP = nucleus pulposus, MRI = magnetic resonance imaging.

**Table 5 medicina-56-00627-t005:** Comparison of imaging findings of patients with leg pain between the G and P groups.

Leg Pain	Group G (*n* = 21)	Group P (*n* = 13)	*p*-Value
disc height decreases ≥ 25%	15	6	0.168 ^†^
HIC of protruded NP on MRI	14	4	**0.042** ^**†**^
the spinal canal occupying ratio ≥ 40%	13	3	**0.028** ^**†**^
Pfirrmann classification			0.591 ^†^
Grade II	7	6	
Grade III	13	7	
Grade IV	1	0	

^†^: χ^2^ test. HIC = high-intensity change, NP = nucleus pulposus, MRI = magnetic resonance imaging. *p* < 0.05 is shown in bold.

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
