# Peer review of "Chemonucleolysis with Chondroitin Sulfate ABC Endolyase for Treating Lumbar Disc Herniation: Exploration of Prognostic Factors for Good or Poor Clinical Outcomes"

_medicina, 2020, doi:10.3390/medicina56110627_

Round 1

Reviewer 1 Report

This clinical study investigated the efficacy of condoliase on the back and leg pain of patients with lumbar disc herniation. The authors retrospectively studied 34 patients who received condoliase injection at a single hospital within one year. Their findings suggested that condoliase improved leg pain scores, especially in younger patients. The effectiveness of condoliase depends on the size and type of protrusion, and the signal intensity of the protruded NP. This study provides some valuable information regarding the patient selection of condoliase injection to treat disc herniation. The manuscript can be improved in writing.

Specific comments:

  1. Abstract: it would be better to mention the retrospective nature of the study.
  2. Fig2: was C used in calculation of the disc height index in this study?
  3. Based on information from Tables 4 and 5, some patients have both leg and back pain. Would there be an outcome difference if a patient has only a leg pain vs a patient has both leg and back pain?
  4. Some results can be reorganized and simplified for easy reading.

Author Response

Ms. Ref. No.:  medicina-956479

Dear Reviewer 1

We thank you for the insightful critique of our manuscript entitled “Chemonucleolysis with chondroitin sulfate ABC endolyase for treating lumbar disc herniation: exploration of prognostic factors for good or poor clinical outcomes.” We appreciate the fact that you have given us the opportunity to submit a revised manuscript and that it will be considered for publication in Medicina.

The manuscript was carefully checked and rewritten according to editor’s and the reviewers’ comments and suggestions. We would like to address those comments point by point, as follows:

â‘ Reviewer #1 suggested to mention the retrospective nature of the study. According to this suggestion, we indicated following sentence at the beginning of Materials & Methods section: Study design: cross-sectional study (highlighted by yellow).

â‘¡Reviewer #1 asked that was C in Figure 2 used in calculation of the disc height index in this study. To avoid misunderstanding of the readers, I modified the text in Materials & Methods section (highlighted by yellow).

â‘¢Reviewer #1 requested to add the comparative data between patients had only a leg pain and patients had both leg and low back pain. Only 2 patients complained leg pain alone. Furthermore, we exclude the patients complained low back pain alone in this study (this is the recommendation from pharmaceutical company developed condoliase). Therefore, the most of the patients complained both leg and low back pain and we could not comparatively analyze patients had only leg pain vs patients had both leg and back pains.

â‘£Reviewer #1 also suggested that some results can be reorganized and simplified for easy reading. This suggestion is quite reasonable, however the improvement of pain to chemonucleolysis were different in part between leg and low back pain. According to this suggestion, we modified table 2-5 and the legends.

Reviewer 2 Report

This manuscript reports the effect of condoliase-based chemonucleolysis on leg and back pain associated to lumbar disc herniation (LDH). Based on changes in the NRS score post-treatment, patients were classified into 2 major groups: good response-G and poor response-P. The results revealed that the treatment was more effective against leg pain. However, there is a tendency for good outcomes in younger patients with back pain as well. Taken together, the study is interesting but has a few limitations:

  • There is a low number of patients and an unequal distribution between males and females.
  • No long term follow up has been presented.
  • The results suggest that younger patients with back pain may respond better than adults. This aspect is more novel and can be further discussed. Is there a rationale to explain such a discrepancy between ages? What proportion of the 9 teenagers was female or male?
  • The authors argue that chemonucleolysis with condoliase is safer and may be used for younger patients. Please expand on the criteria used to evaluate safety following treatment.

Author Response

Ms. Ref. No.:  medicina-956479

Dear Reviewer 2

We thank you for the insightful critique of our manuscript entitled “Chemonucleolysis with chondroitin sulfate ABC endolyase for treating lumbar disc herniation: exploration of prognostic factors for good or poor clinical outcomes.” We appreciate the fact that you have given us the opportunity to submit a revised manuscript and that it will be considered for publication in Medicina.

The manuscript was carefully checked and rewritten according to editor’s and the reviewers’ comments and suggestions. We would like to address those comments point by point, as follows:

â‘ Reviewer #2 expressed concern about limitation of this study (There is a low number of patients and an unequal distribution between males and females. No long term follow up has been presented). According to this concern, we revised study limitations in discussion section “4.1. Limitation of this study”.

â‘¡Reviewer #2 requested to show a rationale to explain different response depending on patient’s age. We speculate the difference might be depending on the water content in nucleus pulposus (the content is predicted more abundant in younger patients than that in elder patients) and the different mechanisms. According to this concern, we modified discussion section.

â‘¢Reviewer #2 also requested to show a proportion (male to female) of the 9 teenagers. According to this concern, we modified discussion section.

â‘£Reviewer #1 asked to expand on the criteria used to evaluate safety following treatment. We have additional data for the safety against disc material and endoplate (we pathologically analyzed disc material after chemonucleolysis). According to this comment, we add additional data (Figure 8) and discussed the safety in Discussion section.

⑤By these revisions, we cited additional references and changed the number.

â‘¥In addition to above reviewer’s comments, we found several minor mistakes in the text. We also corrected these mistakes.
